# Learning Scalable Structural Representations for Link Prediction with Bloom Signatures

## ABSTRACT

Graph neural networks (GNNs) have shown great potential in learning on graphs, but they are known to perform sub-optimally on link prediction tasks. Existing GNNs are primarily designed to learn single-node representations and usually fail to capture pairwise relations between target nodes, which proves to be crucial for link prediction. Recent works resort to learning more expressive edge-wise representations by enhancing vanilla GNNs with structural features such as labeling tricks and link prediction heuristics, but they suffer from high computational complexity and limited scalability. To tackle this issue, we propose to learn structural link representations by augmenting the message passing framework of GNNs with Bloom signatures. Bloom signatures are hashing-based compact encoding of node neighborhoods, which can be efficiently merged to recover various types of edge-wise structural features. We further show that any type of neighborhood overlap-based heuristic can be estimated by a neural network which takes Bloom signatures as input. GNNs with Bloom signatures are provably more expressive than vanilla GNNs and also more scalable than existing edge-wise models. Experimental results on standard link prediction benchmarks show that our proposed model achieves comparable or better performance than existing edge-wise GNN models while being 2-121 × faster and more memory-efficient for online inference.

## CCS CONCEPTS

• **Computing methodologies → Network science; Machine learning**.

## KEYWORDS

Graph Neural Networks, Link Prediction, Graph Representation Learning, Structural Features, Bloom Signatures, Hashing

**ACM Reference Format:**
Anonymous Author(s). 2018. Learning Scalable Structural Representations for Link Prediction with Bloom Signatures. In *Proceedings of Make sure to enter the correct conference title from your rights confirmation email (Conference acronym 'XX)*. ACM, New York, NY, USA, 13 pages. https://doi.org/XXXXXXX.XXXXXXX

## 1 INTRODUCTION

Link prediction is one of the fundamental tasks in graph machine learning and has wide real-world applications in network analysis, recommender systems [3, 38], knowledge graph completion [28, 41], interaction of proteins [27] and drugs [26], and vascular prediction in neuroscience [21]. Early research [20] investigated using hand-crafted score functions to measure the neighborhood similarity between two end nodes for predicting links. These score functions including Common Neighbor (CN) [2], Jaccard index [13], Adamic-Adar (AA) [1] and Resource Allocation (RA) [40] are deterministic and localized, often referred as link prediction heuristics. To incorporate the global structure, node embedding [10, 22] or matrix factorization-based [24] methods are proposed, which map each node in the graph into a low-dimensional hidden space. These vectorized node embeddings preserve certain structural properties of the input graph that can be used for predicting links. Recently, graph neural networks (GNNs) [11, 16] have begun to dominate representation learning on graphs, due to its benefits in combining node features and graph structures through a message passing framework [9]. To predict the target link, GNNs are first applied to the entire graph to obtain the node-wise representation, and then embeddings of two end nodes are aggregated to predict the likelihood of forming a link. GNNs have shown excellent performance on node-level tasks, but sometimes only achieve subpar performance on link prediction, and may even be worse than unsupervised embedding methods or simple heuristics [37, 39].

Multiple studies have revealed the deficiency of directly aggregating node embeddings produced by GNNs as the representation of a link for prediction [6, 19, 25, 31, 37, 39]: (1) It cannot measure the overlap between node neighborhoods as link prediction heuristics do, where [7] proves that GNNs are incapable of counting connected substructures like triangles. (2) It cannot distinguish target nodes that are isomorphic (e.g. nodes $u, w$ are structurally symmetric in Fig. 1) within its $k$-hop induced subgraphs, where [32] proves that the expressive power of GNNs is bounded by the 1-Weisfeiler-Lehman (1-WL) test. For nodes with isomorphic neighborhood, GNNs map them into the same representation due to limited expressiveness, while graph embedding methods do not suffer as their receptive fields are the whole graph. As a node-wise model, GNNs are incapable of producing effective structural representations of target links, which require encoding the pairwise relation between two given candidate nodes.

Recent studies resort to learning more expressive link representations through edge-wise models. Two approaches are used to alleviate the limitations of vanilla GNNs in generating structural link representations. **Coupled**: labeling tricks are a family of hand-crafted and target-link-specific structural features [19, 35, 37, 39, 41]. It injects dependencies between nodes in target links by separately assigning different node labels based on proximity-based metrics within their $k$-hop enclosing subgraphs. Added labels enable the

model to distinguish links containing isomorphic nodes (breaks the symmetry) and to learn data-driven heuristics from labeled neighborhoods. For each target link, it appends node labels after raw features and then runs a GNN to obtain the representation of link-induced subgraph for prediction (often refers as subgraph-based models), which significantly outperforms vanilla GNNs on multiple benchmark datasets but also suffers from high computation overhead. **Decoupled**: directly use heuristic-like pairwise features to generate structural link representations without modifying the input graph, including high-order neighborhood overlap [36], intersection and difference of node neighborhoods [6], and geodesic path between two end nodes [17]). Detaching structural features from message passing achieves better scalability, but this simplification also compromises its capacity and empirical performance.

The emerging challenge in leveraging structural features to yield more expressive link representations lies in the trade-off between feature richness and their computation complexity. Structural link representations is edge conditioned, where the inefficiency and redundancy of materializing structural features for each target link makes it no longer feasible for large-scale graphs; directly using precomputed pairwise features for learning link representations avoids rerunning GNNs and explicitly constructing structural features. Meanwhile, it loses substantial capacity and flexibility for message passing models to capture some important structural signals on the graph, as these features are hand-crafted and collapsed to certain values by fixed score functions (e.g. overlap or difference).

To address the above issues, we propose Bloom Signature, a scalable node-wise neighborhood signature for constructing scalable structural features. It encodes node neighborhoods of different orders into compact bit arrays (termed "signature") through hashing. By design, Bloom Signature can be preprocessed offline in parallel once and efficiently merged to recover edge-wise features online, including a variety of neighborhood overlap-based heuristics. This property avoids expensive overhead of feature materialization for each target link in labeling tricks. Meanwhile, unlike manual pairwise features, it maintains structural signals from node neighborhoods, allowing for the flexibility to learn more expressive link representations and data-driven heuristics. Bloom Signature achieves a balance between model expressiveness and computational complexity for learning structural link representations. Experimental results on 5 public benchmark datasets show that its performance is comparable to or better than existing edge-wise models, achieving $121\times$ speedup over formerly SOTA model SEAL and another $2\times$ speedup over the fastest one.

Our main contributions can be summarized as follows:

- We identify key bottlenecks in applying structural features for learning expressive structural link representations and propose a decoupling mechanism to address the computational challenges of pairwise feature construction while maintaining model expressiveness and empirical performance;
- A scalable hashing-based structural feature, Bloom Signature, is proposed to augment the message passing framework of GNNs, which is a compact encoding of node neighborhoods that can be efficiently merged to recover edge-wise structural features for online training and inference on large-scale graphs.
- We provide error bounds for the estimation of neighborhood intersections from Bloom signatures and further show that any

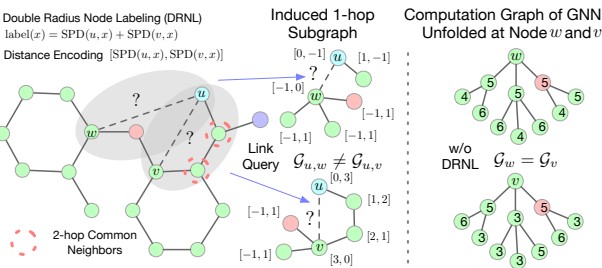

**Figure 1: GNNs cannot correctly predict whether $u$ is more likely linked with $v$ or $w$, because $v$ and $w$ have the same node representation without additional information (colored nodes only, without DRNL labels or encoding of shortest path distances within 3 hops). Representations with distance-based labels added are more expressive to distinguish these two pairs of link $(u, v)$ and $(u, w)$.**

type of overlap-based heuristics can be recovered from neighborhood signatures via neural networks with accuracy guarantee.

## 2 PRELIMINARIES AND RELATED WORK

*Notations.* Let $\mathcal{G} = (\mathcal{V}, \mathcal{E}, X)$ be a undirected graph of $N$ nodes $\mathcal{V} = \{1, 2, \ldots, N\}$ and $E$ edges $\mathcal{E} \subseteq \mathcal{V} \times \mathcal{V}$ with node features of $X \in \mathbb{R}^{N \times F}$. Let $A \in \mathbb{R}^{N \times N}$ be the adjacency matrix of $\mathcal{G}$, where $A_{uv} = 1$ if $(u, v) \in \mathcal{E}$ and $A_{uv} = 0$ otherwise. The degree of node $u$ is $\deg(u) := \sum_{v=1}^{N} A_{uv}$. We denote the $k$-hop neighborhood of node $u$ as $\mathcal{N}^k(u)$, which is the set of all nodes that is connected to $u$ with the shortest path distance $k$. The $k$th-order neighborhood overlap, union and difference between node $u$ and $v$ are given by $\mathcal{N}^k(u) \cap \mathcal{N}^k(v)$, $\mathcal{N}^k(u) \cup \mathcal{N}^k(v)$, and $\mathcal{N}^k(u) - \mathcal{N}^k(v)$. Let $\mathcal{H} : \mathcal{V} \to [n]$ be a pairwise independent hash function, i.e. $\Pr(\mathcal{H}(u) = h_1 \wedge \mathcal{H}(v) = h_2) = 1/n^2$ for $h_1, h_2 \in [n]$, $u \neq v$ and $u, v \in \mathcal{V}$.

*Link Prediction Heuristic.* Common heuristics for link prediction is neighborhood overlap-based with varieties in score functions to measure the similarity. (1) **1-hop neighborhood**: common neighbor (CN) $S_{\text{CN}}(u, v) = \sum_{w \in \mathcal{N}(u) \cap \mathcal{N}(v)} 1$, resource allocation (RA) $S_{\text{RA}}(u, v) = \sum_{w \in \mathcal{N}(u) \cap \mathcal{N}(v)} 1/\deg(w)$, and Adamic-Adar (AA) $S_{\text{AA}}(u, v) = \sum_{w \in \mathcal{N}(u) \cap \mathcal{N}(v)} 1/\log \deg(w)$. (2) **high-order neighborhood**: Neo-GNN [36] utilizes a weighted overlap with node degrees in different orders as $S_{\text{Neo}}(u, v) = \sum_{k_1=1}^{k} \sum_{k_2=1}^{k} (\beta^{k_1+k_2-2} \cdot \sum_{w \in \mathcal{N}^{k_1}(u) \cap \mathcal{N}^{k_2}(v)} f(\deg(w)))$, where $\beta$ is a hyper-parameter and $f$ is a learnable function of degree features. BUDDY [6] uses $k^2$ high-order overlap and $2k$ difference features as $S_{\text{BUDDY}}(u, v) = \{\sum_{w \in \mathcal{N}^{k_1}(u) \cap \mathcal{N}^{k_2}(v)} 1 | k_1, k_2 \in [k]\} || \{\sum_{w \in \mathcal{N}^{k_1}(u) - \cup_{k'=1}^{k} \mathcal{N}^{k'}(v)} 1, \sum_{w \in \mathcal{N}^{k_2}(v) - \cup_{k'=1}^{k} \mathcal{N}^{k'}(u)} 1 | k_1, k_2 \in [k]\}$, where $||$ is concatenation.

*Message Passing Neural Network.* Message passing neural network (MPNN) [9] is a generic framework of GNNs, which learns node representations via iterative aggregations of their local neighborhoods in the graph. To compute the message $\boldsymbol{m}_v^{(l)}$ and the hidden state $\boldsymbol{h}_v^{(l)}$ for each node $v \in \mathcal{V}$ at the $l$-th layer ($l = 1, 2, \ldots, L$):

$$\boldsymbol{m}_v^{(l)} = \text{AGG}\left(\left\{\phi^{(l)}\left(\boldsymbol{h}_v^{(l-1)}, \boldsymbol{h}_w^{(l-1)}\right) | w \in \mathcal{N}(v)\right\}\right) \quad (1)$$

$$\boldsymbol{h}_v^{(l)} = \sigma^{(l)}\left(\boldsymbol{h}_v^{(l-1)}, \boldsymbol{m}_v^{(l)}\right) \quad (2)$$

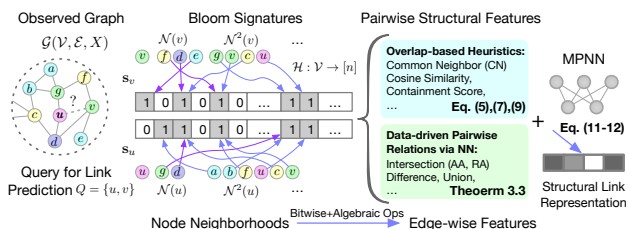

**Figure 2: Bloom Signature: A scalable hashing-based structural feature of node neighborhoods, which can be used to recover neighborhood overlap-based heuristics or learn data-driven pairwise relations for structural link representation.**

where $\phi, \sigma$ are learnable functions of message and update, respectively. AGG is a local permutation-invariant aggregation function (e.g. sum, mean). The initial node representation is set as $\boldsymbol{h}_v^{(0)} = X_v$. The final layer output gives the node representation $\boldsymbol{h}_v$ (omits the superscript ($L$) for simplicity). To predict the likelihood between two candidate nodes $u, v$ to form a link (noted as $\hat{A}_{ij}$) from representations produced by MPNN:

$$\hat{A}_{uv} = \text{sigmoid}(\text{MLP}(\boldsymbol{h}_u \circ \boldsymbol{h}_u)),$$

where $\circ$ is the decoder for link prediction, such as dot product, Hardamard product or concatenation.

**Definition 2.1** (Li et al., Srinivasan and Ribeiro). The *most expressive structural representation of link* gives the same representation if and only if two links are isomorphic (symmetric, on the same orbit) in the graph.

*Structural Features for Link Prediction.* From Def. 2.1, a most expressive structural representation will distinguish all non-isomorphic links, where simply aggregating node representations generated from GNNs fails to do so. GNNs map isomorphic nodes into the same representation as its expressive power is upper-bounded by the 1-WL test. As Fig. 1 shown, such behaviour would lead to two totally unrelated nodes $u, w$ and two neighboring nodes $u, v$ sharing the same link representations, just because the tail nodes $w$ and $v$ have the isomorphic neighborhood. Structure features are introduced to break symmetries in their neighborhood and inject dependency between target end nodes $(u, v)$ through distance-based functions $\text{dis}(w), \forall w \in \mathcal{N}^k(u) \cup \mathcal{N}^k(v)$: Distance Encoding [19] adopts shortest path distance (SPD) or random walk landing probability (RW); Double Radius Node Labeling (DRNL, [37]) $\text{dis}(w) = \text{SPD}(u, w) + \text{SPD}(v, w)$, and Relative Positional Encoding [33, 34] $\text{dis}(w) = \{\text{RW}(u, w) \| \text{RW}(v, w)\}$. It has been proven that a sufficiently expressive GNN with structural features like labeling tricks or link prediction heuristics can learn the most expressive structural representation for links [30].

## 3 BLOOM SIGNATURES FOR STRUCTURAL LINK REPRESENTATION LEARNING

### 3.1 Breakdown Pairwise Structural Features

Using structural features for link representation learning is justified by theoretical foundations and superior empirical performance. However, the successfully deployed structural features are pairwise and edge-specific. Meanwhile, it must be computed before being

**Table 1: Unified Formulation of Node Neighborhood-based Structural Features**

| Feature | Neighborhood | Set operator $\uplus$ | $f(x)$ | $d(\cdot)$ |
|---|---|---|---|---|
| CN | $k = 1$ | $\mathcal{N}(u) \cap \mathcal{N}(v)$ | $x$ | $1$ |
| RA | $k = 1$ | $\mathcal{N}(u) \cap \mathcal{N}(v)$ | $x$ | $1/\deg(w)$ |
| AA | $k = 1$ | $\mathcal{N}(u) \cap \mathcal{N}(v)$ | $x$ | $1/\log \deg(w)$ |
| Neo-GNN | $k_1, k_2 \geq 1$ | $\mathcal{N}^{k_1}(u) \cap \mathcal{N}^{k_2}(v)$ | MLP | $\deg(w)$ |
| BUDDY | $k_1, k_2 \geq 1$ | $\mathcal{N}^{k_1}(u) \cap \mathcal{N}^{k_2}(v)$ $\mathcal{N}^{k_1}(u) - \cup_{k'=1}^{k} \mathcal{N}^{k'}(v)$ | $x$ | $1$ |
| Labeling Trick | $k \geq 1$ | $\mathcal{N}^k(u) \cup \mathcal{N}^k(v)$ | MPNN | $\text{dis}(w)$ |

fed to the model each time during training and inference, which loses the advantage of being easily parallelizable like node features typically processed in GNNs. The challenge of learning structural link representation is how to obtain high quality pairwise structural features at low cost.

By observation, neighborhood intersection-based structural features including common link prediction heuristics, high-order overlap and difference features share the common format as

$$S(u, v) = \sum_{w \in \mathcal{N}^k(u) \uplus \mathcal{N}^k(v)} f(d(w)) \tag{3}$$

We unify them by defining a function $d : \mathcal{V} \rightarrow \mathbb{R}^+$ and a learnable function $f$. Table 1 summarizes the unified formulation for heuristics and high-order pairwise features. Note that, the labeling trick implicitly follows the formulation in Eq. (3) by first labeling nodes via a predefined distance function $\text{dis}(\cdot)$ within the union of two $k$-hop neighborhoods and then feeding those node labels for message passing.

From Eq. (3), node neighborhood is essential to obtaining pairwise structural features, and set operations constitute the main computational cost. To get node labels, one common trick, DRNL requires two traversals over the union neighborhoods, with a complexity of $O(\bar{d}^k)$ where $\bar{d}$ is the average degree. While heuristic-like pairwise features are manually crafted to obtain certain topological statistics (intersection, difference, union) from node neighborhoods. Eliminating node neighborhoods and detach them from message passing do reduce the complexity, but this simplification greatly compromises the model's ability to capture any additional structural signals not included in the computed statistics.

One solution is to decompose structural features into carefully designed node neighborhood representations, so that pairwise features can be efficiently recovered from them online. Such a design brings three main benefits: (1) it maintains the topological information of the neighborhood while providing more richness and flexibility; (2) it decouples the dependence on edges as node-wise representations can be preprocessed offline; (3) there are low-cost substitutes of set operations for reconstructing pairwise features from compact neighborhood representations. However, it is prohibitively expensive to store and operate on the complete node neighborhood, not to mention it is sparse and of varying sizes. An ideal representation of node neighborhood should be space efficient while being able to produce an accurate estimation of pairwise features through lightweight operations during online inference.

### 3.2 Encode Node Neighborhood via Hashing

We propose Bloom Signature, a compact, merge-able node neighborhood encoding, inspired by Bloom filter [4] which is a probabilistic

data structure for set membership testing. Bloom Signature encodes a node's neighborhood as a short-length bit array ("signature") via random hashing without explicitly storing the neighbors. The obtained neighborhood signatures contain provably sufficient structural information to accurately estimate pairwise features, such as intersection and cosine similarity.

For node $u$, its neighborhood $\mathcal{N}(u)$ can be represented by a multi-hot $N$-length binary array $\boldsymbol{u} \in \{0, 1\}^N$, where $\boldsymbol{u}[w] = \mathbb{1}\{w \in \mathcal{N}(u)\}$ (set the $w$-th bit to 1 if node $w$ exists in $\mathcal{N}(u)$ and 0 otherwise). Clearly, such a long and dense representation is impractical, but its sparsity can be exploited to reduce the size of $\boldsymbol{u}$ while maintaining the property of recovering node neighborhood statistics and pairwise structural features. By using a hash function $\mathcal{H}$ to randomly map elements from $\{1, \ldots, N\}$ to $\{1, \ldots, n\}$ where $n \ll N$, the **Bloom Signature** $\boldsymbol{s}_u \in \{0, 1\}^n$ is a compact encoding of the node neighborhood $\mathcal{N}(u)$:

$$s_u[j] = \bigvee_{i:\mathcal{H}(i)=j} \boldsymbol{u}[i], \tag{4}$$

where $\bigvee$ denotes the bit-wise OR operator. $\boldsymbol{s}_u$ refers as the signature of $\mathcal{N}(u)$ with reduced size of $n$ while approximately preserving its neighborhood information. Note that Bloom Signature is a special case of Bloom filters [4] with one hash function. Rather than using it for set membership testing, Bloom Signature encodes node neighborhoods that contain sufficient structural information from which rich features can be constructed and learnt for link prediction. The benefits of Bloom Signature are threefold:

- **Informative**: pairwise heuristics such as neighborhood overlaps can be accurately recovered from Bloom signatures (Sec. 3.3);
- **Flexible**: by feeding Bloom Signature into neural networks, data-driven pairwise features can be learned end-to-end (Sec. 3.4);
- **Expressive**: MPNN is provably more powerful by enhancing it with Bloom Signature (Sec. 3.5).

## 3.3 Recover Pairwise Heuristics from Neighborhood Signatures

After obtaining the Bloom Signature of each node, one can recover neighborhood overlap-based heuristics from a pair of Bloom signatures with accuracy guarantee. This section describes how to reconstruct common neighbors, cosine similarity, and containment score between two neighborhoods by simply merging their corresponding Bloom signatures. We first show how to obtain statistics of node neighborhood $\mathcal{N}_u$ from single signature $\boldsymbol{s}_u$ and then introduce the merge operation of $\boldsymbol{s}_u, \boldsymbol{s}_v$ for pairwise features.

Since the hash function $\mathcal{H}$ is uniformly random, we can establish the relation between the number of non-zeros in $\boldsymbol{s}_u$ and the cardinality of $\mathcal{N}_u$ by $\mathbb{E}(|\boldsymbol{s}_u|/n) = 1 - (1 - 1/n)^{|\mathcal{N}(u)|}$, where $\boldsymbol{s}_u$ has a size of $n$ with $|\boldsymbol{s}_u|$-many non-zero bits. Thus, the size of node $u$'s neighborhood can be estimated from the signature $\boldsymbol{s}_u$ as

$$|\mathcal{N}(u)| \approx \hat{n}_u = \ln\left(1 - \frac{|\boldsymbol{s}_u|}{n}\right)/\ln \rho, \ \rho = 1 - \frac{1}{n} \in (0, 1). \tag{5}$$

The error bound of the above cardinality estimation of $\mathcal{N}_u$ is given by Lemma 3.1, and its detailed proof is provided in Appx. A.2.

**Lemma 3.1.** With probability at least $1 - \delta$, it holds that

$$||\mathcal{N}(u)| - \hat{n}_u| < \sqrt{4m \log \frac{2}{\delta}}, \tag{6}$$

where $m$ is the sparsity of the binary vector $\boldsymbol{u}$ representing $\mathcal{N}(u)$.

Given a pair of $n$-size signatures $\boldsymbol{s}_u, \boldsymbol{s}_v$, their inner-product preserves original neighborhood statistics after random mapping [5]:

$$\mathbb{E}\left(\frac{\langle s_u, s_v \rangle}{n}\right) = (1 - \rho^{|\mathcal{N}(u)|}) - (1 - \rho^{|\mathcal{N}(v)|}) + \rho^{|\mathcal{N}(u)| + |\mathcal{N}(v)| - |\mathcal{N}(u) \cap \mathcal{N}(v)|}.$$

This allows us to express the cardinality (equivalently, common neighbors) of two neighborhood intersection $\mathcal{N}(u) \cap \mathcal{N}(v)$ in terms of the inner-product of two Bloom signatures $\boldsymbol{s}_u, \boldsymbol{s}_v$ as

$$|\mathcal{N}(u) \cap \mathcal{N}(v)| \approx \hat{S}_{\text{CN}}(u, v) = \hat{n}_u + \hat{n}_v - \frac{\ln\left(\rho^{\hat{n}_u} + \rho^{\hat{n}_v} + \frac{\langle s_u, s_v \rangle}{n} - 1\right)}{\ln \rho}, \tag{7}$$

where $\hat{n}_u, \hat{n}_v$ are obtained from Eq. (5) and $\rho = 1 - 1/n$. We further provide the quality analysis of the above intersection estimation and gives the error bound in Theorem 3.2 (detailed in Approx. A.2).

**Theorem 3.2.** With probability at least $1 - 3\delta$, it holds that

$$\left||\mathcal{N}(u) \cap \mathcal{N}(v)| - \hat{S}_{\text{CN}}(u, v)\right| < 6\sqrt{m} + 7\sqrt{2m \ln \frac{2}{\delta}}, \tag{8}$$

where $N$-dim binary vectors of $\mathcal{N}(u)$ and $\mathcal{N}(v)$ have the sparsity at most $m$ with probability at least $1 - \delta'$.

The inner-product of two Bloom signatures can be efficiently obtained by "merging" them in parallel: bit-wise AND operations followed by a summation. From Eqs. (5) and (7), we can accurately estimate the cardinality of a node's neighborhood and pairwise intersections, which forms the basis of neighborhood overlap-based heuristics. Below shows how to generalize this beyond intersection and take into account the neighborhood size of each node. Use cosine similarity $S_{\cos}(u, v) = |\mathcal{N}(u) \cap \mathcal{N}(v)|/\sqrt{|\mathcal{N}(u)||\mathcal{N}(v)|}$ and containment score $S_{\text{cont}}(u, v) = |\mathcal{N}(u) \cap \mathcal{N}(v)|/|\mathcal{N}(u)|$ as an example, we can estimate them from $\hat{n}_u, \hat{n}_v$ and $\hat{S}_{\text{CN}}$ obtained earlier as

$$\hat{S}_{\cos}(u, v) = \frac{\hat{S}_{\text{CN}}(u, v)}{\sqrt{\hat{n}_u \hat{n}_v}}, \qquad \hat{S}_{\text{cont}}(u, v) = \frac{\hat{S}_{\text{CN}}(u, v)}{\hat{n}_u} \tag{9}$$

Note that cosine similarity is symmetric, while containment score is asymmetric, and they both provide fine-level details of the topology in the original node neighborhood. The compactness of Bloom signatures enables us to encode higher-order (i.e. multi-hop) neighborhoods of nodes, which can contain useful features for link prediction models.

## 3.4 Learning Data-driven Pairwise Relations

Common link prediction heuristics can be recovered by simply merging a pair of Bloom signatures. However, most pairwise heuristic are hand-crafted and thus inflexible. The summation in Eq. (3) further collapses the structural information of node neighborhoods into some statistics, making it hardly learnable. Ideally, we would like to capture pairwise relations in an end-to-end manner. For example, SEAL [37] directly modifies the input graph and attaches link-specific distance features for message passing. However, the coupling between node labels and target links makes its deployment extremely expensive in practice. Although some simplifications

**Table 2: Complexity comparison. $h_s$, $h_b$: the complexity of hash operations in Subgraph Sketch and Bloom Signature, respectively. $F$: the dimension of node representations. $k$ is the number of hops for induced subgraphs (node neighborhood). $h$ is the complexity of hashing-based methods to obtain pairwise structural features. When predicting the $q$ target links, time complexity of existing models comes from two parts: message passing and link predictor.**

| Complexity | GNN | SEAL | Neo-GNN | BUDDY | Ours |
|---|---|---|---|---|---|
| Preprocessing | 1 | 1 | 1 | $k\|E\|(\bar{d} + h_s)$ | $kNh_b$ |
| Message Passing | $N\bar{d}F + NF^2$ | 0 | $N\bar{d}F + NF^2$ | $N\bar{d}F$ | $N\bar{d}F + NF^2$ |
| Link Predictor | $qF^2$ | $q(\bar{d}^{k+1}F + \bar{d}^k F^2)$ | $q(\bar{d}^k + F^2)$ | $q(h + F^2)$ | $q(h + F^2)$ |

[17, 33, 34, 36] of labeling tricks have recently been proposed, they still suffer from the high complexity of the set operation ⊎ used to connect neighborhoods of two end nodes.

Bloom signatures encode node neighborhoods in a compact and aligned format that carries provably sufficient structural information to approximately extract pairwise relations without involving any set operations. This property enables neural networks to capture important signals in node neighborhoods and identify data-drive pairwise heuristics from it but at much lower cost. Theorem 3.3 shows that neighborhood overlap-based heuristics can be recovered to arbitrary precision by a neural network (such as MLP) taking a pair of Bloom signatures as input (full proof provided in Appx. A.3).

**Theorem 3.3.** Suppose $S(u, v)$ is a neighborhood intersection-based heuristic as in Eq. (3) with the maximum value of $d(\cdot)$ as $d_{\max} = \max_{w \in \mathcal{V}} d(w)$. Let $p_u = (1 - 1/n)^{|\mathcal{N}(u)|}$ denote the false positive rate of node $u$ for set membership testing in the Bloom signature $\mathbf{s}_u$. Then, there exists an MLP with one hidden layer of width $N$ and ReLU activation, which takes the Bloom signatures of $u$ and $v$ as input and outputs $\hat{S}(u, v) = \text{MLP}(\mathbf{s}_u, \mathbf{s}_v)$, such that with probability 1, $\hat{S}(u, v) - S(u, v) \geq 0$; with probability at least $1 - 3\delta$, it holds that

$$\hat{S}(u, v) - S(u, v) \leq$$
$$d_{\max}\left(\left(1 + \sqrt{\frac{-3\log\delta}{|S_C|p_u p_v}}\right)|S_C|p_u p_v + \left(1 + \sqrt{\frac{-3\log\delta}{|S_{D_v}|p_v}}\right)|S_{D_v}|p_v + \left(1 + \sqrt{\frac{-3\log\delta}{|S_{D_u}|p_u}}\right)|S_{D_u}|p_u\right) \quad (10)$$

where $\mathcal{S}_C = \mathcal{V} \backslash (\mathcal{N}(u) \cup \mathcal{N}(v))$, $\mathcal{S}_{D_v} = \mathcal{N}(u) \backslash \mathcal{N}(v)$, and $\mathcal{S}_{D_u} = \mathcal{N}(v) \backslash \mathcal{N}(u)$.

In fact, the above result can be applied to any pairwise relations of node neighborhoods following the form of Eq. (3), including set difference and union with theorems given in Appx. A.4. The approximation error here is proportional to the false positive rate of Bloom signatures, which means any neighborhood intersection-based heuristic can be exactly recovered if $p_u = 0$. For commonly used heuristics such as CN, AA, and RA, $d_{\max} = 1$.

*Other hashing-based methods.* Subgraph sketch [6] utilizes a combination of MinHash and HyperLogLog to estimate the intersection and complement of high-order node neighborhoods. These hand-crafted pairwise features are used as input to MPNN or MLP for link prediction, obtaining comparable results to labeling tricks. However, subgraph sketch either suffers from high storage and computation overhead (coupled with message passing) or has limited expressiveness (decoupled, only using MLP). The two hashing algorithms

employed require hundreds of repetitions and a large memory budget to achieve reasonable estimation accuracy for pairwise features (see Fig. 3 in Sec. 4.3 for estimation quality study). Moreover, the sketch of subgraphs can only be used to estimate to the statistics of neighborhood intersection and complement, which is inflexible and substantially limits the richness of structural information. In comparison, Bloom Signature captures node membership of encoded neighborhoods, enabling more powerful models to recover any kind of neighborhood overlap-based heuristics (such as RA and AA), or learn data-driven pairwise relations. Both of which are unachievable by subgraph sketch.

## 3.5 Boost Message Passing with Scalable Pairwise Structural Features

Putting all the pieces together, we propose a new scalable MPNN framework augmented by Bloom signatures, which achieves a trade off between efficiency and expressiveness of learning structural link representation. It inherits the rich capacity from pairwise structural features while decoupling from specific edges, enabling online inference. In particularly, we add Bloom signatures as edge feature/weight to augment the message function and/or as additional pairwise features to the link predictor by

$$\mathbf{h}_v^{(l)} = \sigma^{(l)}\left(\mathbf{h}_v^{(l-1)}, \text{AGG}\left(\left\{\phi^{(l)}\left(\mathbf{h}_w^{(l-1)}, \mathbf{h}_v^{(l-1)}, \mathbf{e}_{w,v}\right) | w \in \mathcal{N}(v)\right\}\right)\right), \quad (11)$$

$$\hat{A}_{uv} = \text{sigmoid}\left(\text{MLP}\left(\mathbf{h}_u \circ \mathbf{h}_v || \hat{S}(u, v)\right)\right) \quad (12)$$

where $\mathbf{e}_{w,v} = \psi_f(\{\mathbf{s}_w || \mathbf{s}_v\})$ or $\psi_g(||\mathbf{s}_w - \mathbf{s}_v||)$, and $\psi_f, \psi_g$ are neural encoders such as MLP. $\hat{S}(u, v)$ can be any estimated heuristics or learnable data-driven pairwise relations.

*Expressivity of MPNN with Bloom Signature.* By introducing Bloom Signature, the new framework acts as a more powerful and flexible feature extractor for pairwise relations while enjoying tractable computational complexity of MPNN. As a result, it is more expressive than vanilla GNNs: (1) can measure the overlap between node neighborhoods via recovering intersection-based heuristics from bloom signatures; (2) can avoid node ambiguity via regulated message passing with bloom signatures and recovered pairwise structural features.

*Complexity Comparison.* Table 2 summarizes the computation complexity of vanilla and existing edge-wise GNN models for link prediction. Both BUDDY [6] and Bloom Signature are hashing-based methods, which require preprocessing to obtain $k$-hop subgraph sketches in $O(k|E|h_s)$ and node neighborhood signatures

**Table 3: Summary statistics for evaluation datasets.**

| Dataset | Type | #Nodes | #Edges | $\bar{d}$ | Split(%) |
|---|---|---|---|---|---|
| citation2 | Homo./Social. | 2,927,963 | 30,561,187 | 10.44 | 98/1/1 |
| collab | Homo./Social. | 235,868 | 1,285,465 | 5.45 | 92/4/4 |
| ddi | Homo./Drug | 4,267 | 1,334,889 | 312.84 | 80/10/10 |
| ppa | Homo./Protein | 576,289 | 30,326,273 | 52.62 | 70/20/10 |
| vessel | Homo./Vesicular | 3,538,495 | 5,345,897 | 1.51 | 80/10/10 |

in $O(kNh_b)$. Note that, hashing operations in BUDDY requires hundreds of samples and large storage to generate reasonable estimation, while Bloom Signature only need bit-wise operations and easily parallelizable algebraic operations with accuracy guarantee. Vanilla GNN, Neo-GNN [36] and MPNN with Bloom Signature share the same complexity for message passing. BUDDY precomputes the node feature propagation with $O(k|E|\bar{d})$ and takes another $O(N\bar{d}F)$ to obtain node embeddings. For predicting $q$ target links, vanilla GNN only needs to call MLP with the cost of $O(qF^2)$. SEAL [37] runs MPNN for each link-induced $k$-hop subgraph of size $O(\bar{d}^k)$, resulting in the total complexity of $O(q(\bar{d}^{k+1} + \bar{d}^k F^2))$. Neo-GNN needs additional $O(\bar{d}^k)$ to compute pairwise features before feeding node embeddings into link predictor. To obtain pairwise structural features, both BUDDY and Bloom Signature need extra operations on preprocessed node-wise features with complexity of $h$.

## 4 EVALUATION

In this section, we aim to evaluate the following questions:

- How scalable is Bloom Signature compared to SOTA link prediction models, including hashing-based and other simplification of labeling tricks?
- Can MPNN with Bloom Signature provide prediction performance comparable to existing edge-wise baselines?
- How is estimation quality and efficiency of Bloom Signature?

### 4.1 Experiment Setup

**Datasets** Table 3 summarizes the statistics of datasets used to benchmark different models for link prediction. Five homogeneous networks are selected from the Open Graph Benchmark (OGB) [12] at different scales (4K ~ 3.5M nodes and 1.2M ~ 30.6M edges), various densities (average degree $\bar{d}$ from 1.51 to 312.84), and with distinguishing characteristics: popular social networks of citation - citation2 and collaboration - collab; biological network of protein interactions - ppa, drug interactions - ddi and brain vessels - vessel. Social networks play a key role in network analysis of modeling real-word dynamics. Recently, biological networks are emerging a new data source of network science research. Particularly, understanding the interaction between proteins, drugs and the structure of brain vessel network are of unique significance for scientific discovery [14], which can be used for new drug discovery and early disease detection of neurological disorders.

**Baselines** We consider both classic approaches and state-of-the-art GNN-based models: (1) **Link Prediction Heuristics**: Common Neighbors (CN) [2], Adamic-Adar (AA) [1], and Resource Allocation (RA) [40]. (2) **Embedding Methods**: Matrix Factorization (MF) [24], Multi-layer Perceptron (MLP) and Node2Vec [10]. (3) **Vanilla GNNs**: Graph Convolutional Network (GCN) [16], GraphSAGE

[11] and Graph Attention Network (GAT) [29]. **Edge-wise GNNs**: SEAL [37, 39], Neo-GNN [36], GDGNN [17], BUDDY [6].

**Settings** Data split of OGB is used to isolate validation and test links from the input graph. We adopt official implementations of all baselines with tuned parameters that match their reported results as [18]. All experiments are run 10 times independently, and we report the mean performance and standard deviation.

**Evaluation Metrics** Ranking-based metrics (i.e. mean reciprocal rank (MRR) and Hits@K, $K \in \{20, 50, 100\}$) and the area under the curve (AUC) are used for standard benchmark evaluation.

**Environment** We use a server with two Intel Xeon Gold 6248R CPUs, 1TB DRAM, and two NVIDIA A100 (80GB) GPUs (only one GPU is used per model). Codebase is built on PyTorch 1.12, PyG 2.3, DGL 1.0.2, and numba 0.56.

### 4.2 Prediction Performance Analysis

Table 4 shows the prediction performance of different methods. On these five link prediction benchmarks, edge-wise models significantly outperform both vanilla GNNs and embedding-based models, especially for two challenging biological networks ppa and vessel. Link prediction in biological networks relies on pairwise structural information that vanilla GNNs have limited expressiveness to capture. Among edge-wise models, MPNN with Bloom Signature achieves comparable or better performance than BUDDY (hashing-based) and other simplified labeling tricks including Neo-GNN and GDGNN, and consistently outperforms SEAL (formerly SOTA), which validates the effectiveness of our proposed compact neighborhood encoding. Note that unlike BUDDY which explicitly rely on manual features such as CN and RA, our model can capture richer and complex pairwise structural relations from Bloom Signature, which results in performance exceeding common link prediction heuristics on all five datasets.

### 4.3 Quality Analysis of Hashing-based Estimation for Pairwise Heuristics

To measure the quality of hashing-based methods, Bloom Signature and Subgraph Sketch are compared on a variety of randomly generated graphs and real-world networks for estimating pairwise structural features. The key metric is the trade-off between estimation accuracy and memory cost of hashing. We choose the cardinality of 1-hop and 2-hop neighborhood intersection as the estimated statistic, since it is essential for many commonly used heuristics and pairwise structural features as discussed in Sec. 3.

*Random Graphs.* Two types of representative random network models are picked: Erdős–Rényi [8] and Barabási–Albert [2]. Each random graph is generated with 10,000 nodes. In Erdős–Rényi model, each edge is drawn from the binomial distribution with probability $p$. In Barabási–Albert, each node is added incrementally, with $m$ new edges attached to existing nodes with preferential attachment, meaning the sampling probability is proportional to the node degree. Graphs drawn from Barabási–Albert model have power-law distributed node degrees, which reflects the structure of real-world networks such as the world-wide web, social networks, and academic graphs.

Fig. 3 (UP) presents the mean absolute error (MAE) of intersection estimation for two hashing-based methods on Erdős–Rényi

**Table 4: Results (%) on OGB datasets for link prediction. Highlighted and underlined are the results ranked FIRST, second.**

| | Models | ogbl-collab Hits@50 | ogbl-ddi Hits@20 | ogbl-ppa Hits@100 | ogbl-citation2 MRR | ogbl-vessel AUC |
|---|---|---|---|---|---|---|
| Heuristic | CN | 61.37 | 17.73 | 27.65 | 50.31 | 48.49 |
| | AA | 64.17 | 18.61 | 32.45 | 51.69 | 48.49 |
| | RA | 63.81 | 6.23 | 49.33 | 51.65 | 48.49 |
| Embedding | MF | 41.81 ± 1.67 | 23.50 ± 5.35 | 28.40 ± 4.62 | 50.57 ± 12.14 | 49.97 ± 0.05 |
| | MLP | 35.81 ± 1.08 | N/A | 0.45 ± 0.04 | 38.07 ± 0.09 | 50.28 ± 0.00 |
| | Node2Vec | 49.06 ± 1.04 | 34.69 ± 2.90 | 26.24 ± 0.96 | 45.04 ± 0.10 | 47.94 ± 1.33 |
| Vanilla | GCN | 54.96 ± 3.18 | 49.90 ± 7.23 | 29.57 ± 2.90 | 84.85 ± 0.07 | 43.53 ± 9.61 |
| | SAGE | 59.44 ± 1.37 | 49.84 ± 15.56 | 41.02 ± 1.94 | 83.06 ± 0.09 | 49.89 ± 6.78 |
| | GAT | 55.00 ± 3.28 | 31.88 ± 8.83 | OOM | OOM | OOM |
| Edge-wise | SEAL | 63.37 ± 0.69 | 25.25 ± 3.90 | 48.80 ± 5.61 | 86.93 ± 0.43 | **80.50 ± 0.21** |
| | Neo-GNN | **66.13 ± 0.61** | 20.95 ± 6.03 | 48.45 ± 1.01 | 83.54 ± 0.32 | OOM |
| | GDGNN | 54.74 ± 0.48 | 21.01 ± 2.09 | 45.92 ± 2.14 | 86.96 ± 0.28 | 75.84 ± 0.08 |
| | BUDDY | 64.59 ± 0.46 | 29.60 ± 4.75 | 47.33 ± 1.96 | **87.86 ± 0.18** | 65.30 ± 0.09 |
| | Ours | 65.65 ± 0.32 | **54.33 ± 14.94** | **49.80 ± 1.74** | 87.29 ± 0.20 | 79.83 ± 0.52 |

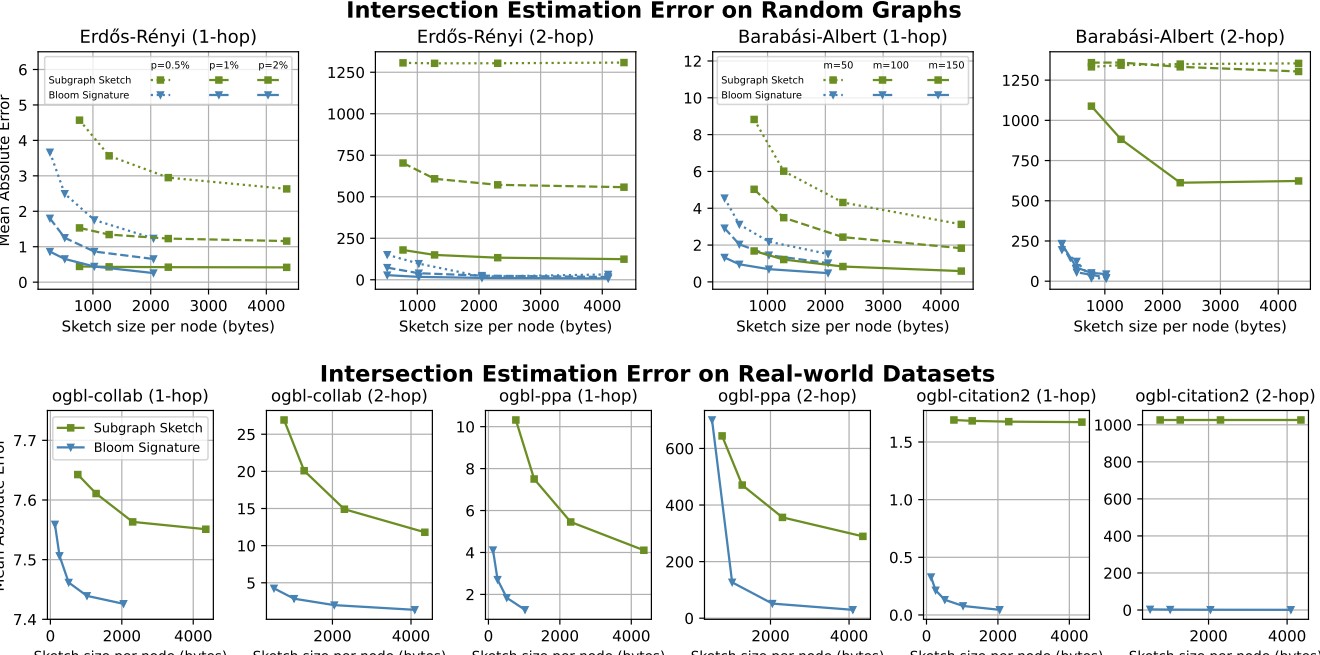

**Figure 3: Mean absolute errors (MAE) of cardinality estimation for 1-hop and 2-hop neighborhood intersection on random networks and real-world networks by Bloom Signature and subgraph sketch with varying memory budget. Under the same memory budget, Bloom Signature produces estimations with up to 2.83× and 68.52× lower error than subgraph sketch for 1-hop and 2-hop neighborhood intersection on random graphs, and with up to 12.84× and 303.10× lower error on real graphs.**

($p \in \{0.5\%, 1\%, 2\%\}$) and Barabási–Albert ($m \in \{50, 100, 150\}$) models with varying density and degree distribution. Under the same memory budget of hashing, Bloom Signature achieves lower estimation error than Subgraph Sketch in almost all cases. As the number of hops increases, the estimation quality of Subgraph Sketch degrades significantly. For 2-hop neighborhood intersection, the estimation error of Subgraph Sketch levels off quickly and does not

approach zero even with increasing memory budget. On the other hand, Bloom Signature is capable of achieve low estimation error for high-order neighborhoods, and the estimation error approaches zero as memory budget increases. Bloom Signature produces estimations of 1-hop neighborhood intersection with up to 2.83× lower error than Subgraph Sketch under the same memory budget, and with up to 68.52× lower error for 2-hop estimations.

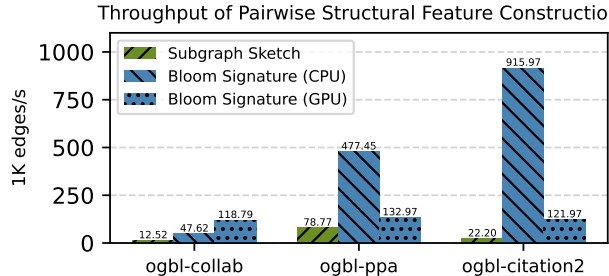

Figure 4: Throughput of constructing pairwise structural features by Subgraph Sketch and Bloom Signature on three OGB datasets. Bloom Signature achieves up to 41.25× and 9.48× higher throughput than Subgraph Sketch on CPU (same number of threads) and GPU, respectively.

Table 5: Runtime Comparison. The row of Train records the time per 10K edges, and Inference is the full test set.

| dataset | time (s) | SEAL | GDGNN | BUDDY | GCN | Ours |
|---------|----------|------|-------|-------|-----|------|
| citation2 | Prep. | 0 | 338 | 398 | 17 | 92 |
| | Train | 3.52 | 2.26 | 0.11 | 0.13 | 0.20 |
| | Inf. | 24,626 | 5,460 | 128 | 15 | 204 |

*Real-word Networks.* The MAE comparison of intersection estimation on three real-world networks are shown in the second half of Fig. 3. Bloom Signature consistently produces more accurate results thanks to the design of compact bit encoding and non-sampling based estimation from Eq. (5). Under the same memory budget, Bloom Signature produces with up to 12.84× and 303.10× lower error than Subgraph Sketch for estimating 1-hop and 2-hop neighborhood intersection, respectively. Note that the estimation quality is dependent upon the degree distribution of the graphs and the actual intersection size, as larger neighborhoods are more difficult to encode and more collisions in encoding are occurred. Aligning with Theorem 3.2, Bloom Signature performs uniformly better than Subgraph Sketch on all real-world networks at different memory budget. This observation is also consistent with the performance difference between BUDDY and our model in Table 4, when estimated pairwise structural features are used for prediction.

## 4.4 Efficiency and Scalability Analysis

Figure 4 compares the efficiency of two hashing-based methods by measuring the throughput of computing 2-hop pairwise structural features. Subgraph Sketch is currently only deployed on CPU, while Bloom Signature can perform accelerated calculations on CPU and GPU. Bloom Signature has 41.25× and 9.48× higher throughput than Subgraph Sketch on CPU and GPU, respectively. Bloom Signature uses only lightweight bitwise and algebraic operations, which favors multithreading on CPUs with lower I/O overhead. Running on GPU introduces I/O inefficiencies, especially on large-scale graphs (e.g. `citation2`). This hurts hardware-accelerated performance and results in Bloom Signature having relatively lower throughput on GPU than on CPU.

Table 5 lists the end-to-end runtime of edge-wise GNN models on the largest benchmark graph `citation2`. The dynamic mode of SEAL (online subgraph extraction) is used due to its extremely high

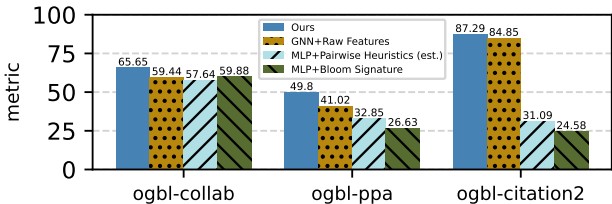

Figure 5: Comparison of different model configurations of MPNN with Bloom Signature on three OGB datasets.

time and space complexity on large graphs. Similarly, only numbers of BUDDY is reported as its message passing version ELPH [6] is infeasible on `citation2`. We also list GCN as a reference point. We omit the result of Neo-GNN as it is even slower than SEAL even only runs MPNN once but needs $k$-order pairwise features. BUDDY precomputes the node feature propagation with sketches and only adopts simplified message passing (shown in Table 2), which is reflected in high cost of preprocessing and low training latency. GDGNN requires to find the geodesic path between node pairs first and then runs MPNN. Bloom Signature only needs to be computed once for $k$-hop node neighborhoods. Overall, MPNN with Bloom Signature does not suffer from high complexity of pairwise structural features as SEAL, GDGNN and Neo-GNN and is orders of magnitude faster. In particular, Bloom Signature is 17.6–120.7 × faster than SEAL in dynamic mode for training and inference on `citation2` and is overall 2× faster than BUDDY (under fair comparison on preprocessing of hashing and full inference with pairwise structural features).

## 4.5 Ablation Study

To validate the design of Bloom Signature, we conduct an ablation analysis and report results on three OGB datasets with the following configurations in Fig. 5: (1) vanilla GNN with only raw node features; (2) MLP with multi-hop estimated pairwise structural features from Bloom Signature; and (3) MLP with multi-hop raw pairwise Bloom Signatures as input. Neither MPNN with raw node features nor MLP with estimated/raw pairwise structural features alone can achieve the best performance. Note that, MLP taking a pair of raw Bloom Signatures as input is comparable or outperforms MLP using estimated pairwise structural features, indicating that Bloom Signature encodes sufficient structural information of node neighborhood for model to capture beyond handcrafted heuristics.

## 5 CONCLUSION

In this work, we present a new MPNN framework with scalable structural features for structural link representation learning, which is based on the observation and analysis of key bottlenecks in the most expressive edge-wise GNN models. A novel efficient structural feature termed Bloom Signature is proposed to encode node neighborhoods through hashing, which is decoupled from specific edges. MPNN with Bloom Signatures is provably more expressive than vanilla GNNs and also more scalable than existing edge-wise models. In practice, it achieves much better time and space complexity and superior predictive performance on a range of standard link prediction benchmarks.

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

# A APPENDIX

## A.1 Experiment Details

*Model Architecture.* We use 3 layers of either GCN or GraphSAGE (the one performs the best) as the backbone of MPNN in Eq. (11) with extended support of edge features. 3-layer MLP of hidden dimension 256 with layer normalization is used for link predictor (dot/diff) and pairwise structural feature encoder. Bloom Signatures of 2-hop node neighborhoods are used for all five datasets, with sizes of $[1024, 4096]$ or $[2048, 8192]$.

*Model Hyperparameter.* We perform heuristic guided searches for tuning. Hyperparameters were selected to maximize metrics on the validation set. The best hyperparameters picked for each model can be found in our code. We set the ratio between the output of MPNN and structural feature encoder as a learnable parameter.

*Training Process.* We use Adam [15] as the optimizer to optimize model parameters with learning rate $0.0005 - 0.01$ and set the maximum epoch of 500. Each positive edge is paired with a random picked negative edge, except for `vessel`. Pairwise structural features are computed on the fly from a pair of Bloom Signatures.

## A.2 Error Bounds for Pairwise Structural Feature Estimation from Bloom Signatures

Given a $N$-dimensional binary vector $x \in \{0, 1\}^N$, Bloom Signature reduces it to a $n$-dimensional binary vector $s_x \in \{0, 1\}^n$. Let $N = \tau m \sqrt{m \ln\left(\frac{2}{\delta}\right)}$, and $m$ is non-zero value (sparsity) of $x$ with probability at least $1 - \delta'$. It randomly maps each bit position $\{i\}_{i=1}^N$ to an integer $\{j\}_{j=1}^n$ as shown in Eq. (4). To compute the $j$-th bit of $s_x$, it checks which bit positions have been mapped to $j$, computes the `bitwise - OR` of the bits located at those positions and assigns it to $s_x[j]$.

**Lemma A.1.**

$$\mathbb{E}\left[\frac{|s_x|}{N}\right] = (1 - \rho^{|x|}), \text{ where } \rho = 1 - \frac{1}{n} \in (0, 1), \quad (13)$$

$$\mathbb{E}\left[\langle s_x, x_y \rangle / n\right] = (1 - \rho^{|x|})(1 - \rho^{|y|}) + \rho^{|x|+|y|}\left[\left(\frac{1}{\rho}\right)^{\langle x, y \rangle} - 1\right]$$

$$= 1 - \rho^{|x|} - \rho^{|y|} + \rho^{|x|+|y|-\langle x, y \rangle} \quad (14)$$

**Lemma A.2.** Given $n \geq cm^{\frac{3}{2}}$, with probability at least $1 - \delta$, it holds that

$$||s_x| - \mathbb{E}[|s_x|]| < \frac{1 + \sqrt{(c^2 + c) \log \frac{2}{\delta}}}{c^2} \sqrt{m}.$$

PROOF. Given a measurable space $(\Omega, \mathcal{F})$ with $\Omega = \{0, 1, ..., n\}$ and corresponding $\sigma$-algebra $\mathcal{F}$, consider a sequence of random variables $X_1, X_2, ..., X_m$. Here, $X_k$ represents the count of occupied bins after $k$ ball throws. As the processes are independent, the probability for the $k + 1$-th ball to land in an empty bin is $1 - \frac{x_k}{n}$. This implies $X_{k+1} = X_k + Z_{k+1}$, where $Z_{k+1}$ follows a Bernoulli distribution with parameter $1 - \frac{x_k}{n}$ (noted as $\text{Ber}(1 - \frac{x_k}{n})$), and we

initialize with $X_0 = 0$. Further, define $Y_k = \frac{x_k - n}{(1 - \frac{1}{n})^{k-1}}$ and $Y_1 = 1 - n$, we observe that

$$\mathbb{E}(Y_{k+1}|\sigma(Z_1, ..., Z_k)) = \frac{X_k - n}{(1 - \frac{1}{n})^k} + \frac{1 - \frac{x_k}{n}}{(1 - \frac{1}{n})^k} + \mathbb{E}\left(\frac{Z_{k+1}}{(1 - \frac{1}{n})^k}\right)$$

$$= \frac{(1 - \frac{1}{n})x_k + 1 - n}{(1 - \frac{1}{n})^k} = \frac{x_k - n}{(1 - \frac{1}{n})^{k-1}} = Y_k \quad (15)$$

Thus, $(Y_{k+1})$ form a martingale adapted to filteration $(\sigma(Z_1, ..., Z_k))$. From Azuma Hoeffding inequality, we have

$$\mathbb{P}[|Y_m - Y_1| \geq \epsilon] \leq 2 \exp\left(\frac{-\epsilon^2}{2 \sum_{k=1}^{m-1} c_k^2}\right) \quad (16)$$

where $c_k = 1/(1 - \frac{1}{n})^{k-1} \geq |Y_{k+1} - Y_k|$.

Consider $\sum_{k=1}^{m-1} c_k^2$,

$$\sum_{k=1}^{m-1} c_k^2 = \sum_{k=1}^{m-1} \frac{1}{(1 - \frac{1}{n})^{k-1}} \leq \int_0^{m-1} e^{x \log\left(\frac{1}{1 - \frac{1}{n}}\right)} dx \quad (17)$$

$$= \frac{1}{\log(1 + \frac{1}{n-1})} \cdot \left[(1 + \frac{1}{n-1})^{m-1} - 1\right] \quad (18)$$

$$\overset{(i)}{\leq} (n - \frac{1}{2})\left[(1 + \frac{1}{n-1})^{m-1} - 1\right] \quad (19)$$

where $(i)$ is from $\frac{2x}{2+x} \leq \log(1 + x), \forall x \geq 0$.

Let $m = O_n(n^{\frac{2}{3}})$, specifically, $n \geq cm^{\frac{3}{2}}$, to have the concentration hold for $Y_k$, we need to make sure that

$$2 \exp\left(\frac{-\epsilon^2}{2 \sum_{k=1}^{m-1} c_k^2}\right) \leq \delta \quad (20)$$

$$\Leftrightarrow \frac{\epsilon^2}{2 \sum_{k=0}^{m-1} c_k^2} \geq \log \frac{2}{\delta} \quad (21)$$

$$\Leftarrow \epsilon^2 \geq \left(\left(1 + \frac{1}{n-1}\right)^{(m-1)} - 1\right)(n - \frac{1}{2}) \log \frac{2}{\delta} \quad (22)$$

$$\Leftarrow \epsilon^2 \geq (m + \frac{1}{c}\sqrt{m} + o(\sqrt{m})) \log \frac{2}{\delta} \Leftarrow \epsilon \geq \sqrt{(1 + \frac{1}{c})m \log \frac{2}{\delta}} \quad (23)$$

From the concentration of $Y_k$, we further derive the concentration property of the number of non-empty bins, i.e. $X_m - X_0$. Since

$|Y_m - (1 - n)| \leq \epsilon$ w.h.p. $Y_m \sim (1 - n) \pm \epsilon$, we have

$$|X_m - X_0 - m| \tag{24}$$

$$= |(1 - \frac{1}{n})^{m-1} Y_m + n - m| \tag{25}$$

$$= |(1 - \frac{1}{n})^{m-1} [(1 - n) \pm \epsilon] + n - m| \tag{26}$$

$$= |(1 - \frac{m}{n}) - (1 - \frac{1}{n})^m (1 \pm \frac{\epsilon}{n-1}))| \cdot n \tag{27}$$

$$= |1 - \frac{m}{n} - (1 - \frac{m}{n} + \frac{m^2}{2n^2})(1 \pm \frac{(\sqrt{(1 + \frac{1}{c}) \log \frac{2}{\delta}}) \sqrt{m}}{n-1}) + o(\frac{\sqrt{m}}{n})| \cdot n \tag{28}$$

$$\leq (\frac{m^2}{2n^2} + (\sqrt{(1 + \frac{1}{c}) \log \frac{2}{\delta}}) \frac{\sqrt{m}}{n}) \cdot n \leq \frac{1 + \sqrt{(4c^2 + 4c) \log \frac{2}{\delta}}}{2c^2} \sqrt{m} \tag{29}$$

□

**Lemma A.3.** With probability at least $1 - \delta$, it holds that

$$|\langle s_u, s_v \rangle - \mathbb{E}[\langle s_u, s_v \rangle]| < \sqrt{2 \min\{|\mathcal{N}(u)|, |\mathcal{N}(v)|\} \log \frac{2}{\delta}}$$

Proof. For any given $\mathcal{N}(u), \mathcal{N}(v) \in \{0, 1\}^N$, we partition $\{1, 2, ..., N\}$ in to four sets: (a) $A = \{j | \mathcal{N}(u)[j] = \mathcal{N}(v)[j] = 1\}$; (b) $B = \{j | \mathcal{N}(u)[j] = 1, \mathcal{N}(v)[j] = 0\}$; (c) $C = \{j | \mathcal{N}(u)[j] = 0, \mathcal{N}(v)[j] = 1\}$; (d) $D = \{j | \mathcal{N}(u)[j] = \mathcal{N}(v)[j] = 0\}$. The random mapping can be viewed as throwing different balls in to $n$ bins. Let red balls denote indices in $A$, blue balls and green balls denote indices in $B$ and $C$ respectively. We say a bin is non-empty if there exist at least one red ball or same amount of blue and green balls. WLOG, we can assume $|B| > |C|$, we further simplified the process as first we throw $|B|$ blue balls. After fixing these balls, we further throw $|A| + |C|$ balls. Given measurable space $(\Omega, \mathcal{F})$ with $\Omega = \{0, 1, ..., n\}$ and corresponding $\sigma$-algebra $\mathcal{F}$, define a sequence of random variables $X_1, X_2, ..., X_{|A|+|C|}$ where $X_k$ represents the count of occupied bins after $k$ ball throws. Since we have Lipschitz condition $|X_{k+1} - X_k| \leq 1$, and from Azuma-Hoeffding inequality

$$\mathbb{P}(|\langle s_u, s_v \rangle - \mathbb{E}[\langle s_u, s_v \rangle]| \geq \epsilon) \leq 2 \exp(\frac{-\epsilon^2}{2(|A| + |C|)}) \tag{30}$$

Further let $\delta \geq 2 \exp(\frac{-\epsilon^2}{2(|A|+|C|)})$, we have

$$\epsilon \geq \sqrt{2(|A| + |C|) \log \frac{2}{\delta}} \tag{31}$$

Since $|A| + |C| = \min\{|\mathcal{N}(u)|, |\mathcal{N}(v)|\}$, we have we probability at least $1 - \delta$,

$$|\langle s_u, s_v \rangle - \mathbb{E}[\langle s_u, s_v \rangle]| \leq \sqrt{2 \min\{|\mathcal{N}(u)|, |\mathcal{N}(v)|\} \log \frac{2}{\delta}} \tag{32}$$

□

**Lemma A.4.** With probability at least $1 - \delta$, it holds that

$$||\mathcal{N}(u)| - \hat{n}_u| < (\frac{1}{2c^2 \sqrt{\log \frac{2}{\delta}}} + \frac{\sqrt{c^2 + c}}{c^2}) \cdot \sqrt{m \log \frac{2}{\delta}}$$

Proof. From Lemma A.1 and Eq. (5), we have $\rho^{|\mathcal{N}(u)|} - \rho^{\hat{n}_u} = \frac{|s_u| - \mathbb{E}[|s_u|]}{n}$. Let $\gamma = \frac{1 + \sqrt{(4c^2 + 4c) \log \frac{2}{\delta}}}{2c^2}$. From Lemma A.2, with probability at least $1 - \delta$, we have

$$|\rho^{|\mathcal{N}(u)|} - \rho^{\hat{n}_u}| = |\frac{|s_u| - \mathbb{E}[|s_u|]}{n}| \tag{33}$$

$$\leq \frac{\gamma \cdot \sqrt{m}}{n} = \frac{1}{m}(\frac{1}{2c^2 \tau \sqrt{\log \frac{2}{\delta}}} + \frac{\sqrt{4c^2 + 4c}}{2c^2 \tau}) \tag{34}$$

From Pratap et al. [23], we have

$$\frac{1}{2}(1 - \rho^{||\mathcal{N}(u)| - \hat{n}_u|}) \leq \frac{1}{m}\left(\frac{1}{2c^2 \tau \sqrt{\log \frac{2}{\delta}}} + \frac{\sqrt{4c^2 + 4c}}{2c^2 \tau}\right) \tag{35}$$

$$\Leftrightarrow ||\mathcal{N}(u)| - \hat{n}_u| \log \rho \geq \log\left(1 - \frac{2}{m}\left(\frac{1}{2c^2 \tau \sqrt{\log \frac{2}{\delta}}} + \frac{\sqrt{4c^2 + 4c}}{2c^2 \tau}\right)\right) \tag{36}$$

$$\Leftarrow ||\mathcal{N}(u)| - \hat{n}_u| \leq \frac{2(\frac{1}{2c^2 \tau \sqrt{\log \frac{2}{\delta}}} + \frac{\sqrt{4c^2 + 4c}}{2c^2 \tau})}{m - 2(\frac{1}{2c^2 \tau \sqrt{\log \frac{2}{\delta}}} + \frac{\sqrt{4c^2 + 4c}}{2c^2 \tau})} \cdot \frac{1}{\log \frac{1}{\rho}} \tag{37}$$

$$\Leftarrow ||\mathcal{N}(u)| - \hat{n}_u| < \frac{(\frac{1}{2c^2 \tau \sqrt{\log \frac{2}{\delta}}} + \frac{\sqrt{4c^2 + 4c}}{2c^2 \tau})}{m - 2(\frac{1}{2c^2 \tau \sqrt{\log \frac{2}{\delta}}} + \frac{\sqrt{4c^2 + 4c}}{2c^2 \tau})} \cdot \tau m \sqrt{m \log \frac{2}{\delta}} \tag{38}$$

$$\Leftarrow ||\mathcal{N}(u)| - \hat{n}_u| < (\frac{1}{2c^2 \tau \sqrt{\log \frac{2}{\delta}}} + \frac{\sqrt{4c^2 + 4c}}{2c^2 \tau}) \cdot \tau \sqrt{m \log \frac{2}{\delta}} \tag{39}$$

□

To estimate the size of intersection $\mathcal{N}(u) \cap \mathcal{N}(v)$, we can use their Bloom signatures of $s_u, s_v$ as follows

$$\hat{S}(u, v) = \hat{n}_u + \hat{n}_v - \frac{\ln\left(\rho^{\hat{n}_u} + \rho^{\hat{n}_v} + \frac{\langle s_u, s_v \rangle}{n} - 1\right)}{\ln \rho}$$

where $\hat{n}_u = \ln(1 - |s_u|/n)/\ln \rho$ and $\hat{n}_v = \ln(1 - |s_v|/n)/\ln \rho$.

**Lemma A.5.** With probability at least $1 - 3\delta$, it holds that

$$\left||\mathcal{N}(u) \cap \mathcal{N}(v)| - \hat{S}(u, v)\right| < \frac{6\sqrt{m}}{c^2} + (5\sqrt{\frac{c+1}{c^3}} + 2\sqrt{2})\sqrt{m \log \frac{2}{\delta}},$$

where $m = \max(|u|, |v|)$.

Proof. From Lemma A.1, we have

$$\langle u, v \rangle = |u| + |v| - \ln\left(\rho^{|u|} + \rho^{|v|} + \mathbb{E}[\langle s_u, s_v \rangle]/n - 1\right)/\ln \rho$$
$$\hat{S}(u, v) = \hat{n}_u + \hat{n}_v - \ln\left(\rho^{\hat{n}_u} + \rho^{\hat{n}_v} + \langle s_u, s_v \rangle/n - 1\right)/\ln \rho \tag{40}$$

in which $|u| \approx \hat{n}_u$, $|v| \approx \hat{n}_v$ (Lemma A.4), and $\mathbb{E}[\langle s_u, s_v \rangle] \approx \langle s_u, s_v \rangle$ (Lemma A.3) with probability at least $1 - \delta$. In Lemma A.4, we show that

$$|\hat{n}_u - |u|| < (\frac{1}{2c^2 \sqrt{\log \frac{2}{\delta}}} + \frac{\sqrt{c^2 + c}}{c^2}) \cdot \sqrt{m \log \frac{2}{\delta}} \tag{41}$$

$$|\hat{n}_v - |v|| < \left(\frac{1}{2c^2\sqrt{\log\frac{2}{\delta}}} + \frac{\sqrt{c^2+c}}{c^2}\right) \cdot \sqrt{m\log\frac{2}{\delta}} \tag{42}$$

We have

$$|\hat{S}(u,v) - \langle \boldsymbol{u}, \boldsymbol{v}\rangle| \tag{43}$$

$$\leq |\hat{n}_u + \hat{n}_v - \ln\left(\rho^{\hat{n}_u} + \rho^{\hat{n}_v} + \langle s_u, s_v\rangle/n - 1\right)/\ln\rho \tag{44}$$

$$- (|\boldsymbol{u}| + |\boldsymbol{v}| - \ln\left(\rho^{|\boldsymbol{u}|} + \rho^{|\boldsymbol{v}|} + \mathbb{E}\left[\langle s_u, s_v\rangle\right]/n - 1\right)/\ln\rho|) \tag{45}$$

$$\overset{(i)}{\leq} \frac{2\sqrt{m}}{c^2} + \sqrt{\frac{c+1}{c^3}}\sqrt{m\log\frac{2}{\delta}}$$

$$+ \frac{1}{\ln\rho}|\ln\frac{\rho^{|\boldsymbol{u}|} + \rho^{|\boldsymbol{v}|} + \mathbb{E}\left[\langle s_u, s_v\rangle\right]/n - 1}{\rho^{\hat{n}_u} + \rho^{\hat{n}_v} + \langle s_u, s_v\rangle/n - 1}| \tag{46}$$

where $(i)$ is from Lemma A.4.

Further,

$$\frac{1}{\ln\rho}\left|\ln\frac{\rho^{|\boldsymbol{u}|} + \rho^{|\boldsymbol{v}|} + \mathbb{E}\left[\langle s_u, s_v\rangle\right]/n - 1}{\rho^{\hat{n}_u} + \rho^{\hat{n}_v} + \langle s_u, s_v\rangle/n - 1}\right| \tag{47}$$

$$\leq \frac{1}{\ln\rho} \cdot$$

$$\frac{|\rho^{|\boldsymbol{u}|} + \rho^{|\boldsymbol{v}|} + \mathbb{E}\left[\langle s_u, s_v\rangle\right]/n - \rho^{\hat{n}_u} + \rho^{\hat{n}_v} + \langle s_u, s_v\rangle/n|}{\max\{\rho^{|\boldsymbol{u}|} + \rho^{|\boldsymbol{v}|} + \mathbb{E}\left[\langle s_u, s_v\rangle\right]/n - 1, \rho^{\hat{n}_u} + \rho^{\hat{n}_v} + \langle s_u, s_v\rangle/n - 1\}} \tag{48}$$

$$\leq \frac{\frac{2\sqrt{m}}{c^2} + 2\sqrt{\frac{c+1}{c^3}m\log\frac{2}{\delta}} + \sqrt{2m\log\frac{2}{\delta}}}{\max\{\rho^{|\boldsymbol{u}|} + \rho^{|\boldsymbol{v}|} + \mathbb{E}\left[\langle s_u, s_v\rangle\right]/n - 1, \rho^{\hat{n}_u} + \rho^{\hat{n}_v} + \langle s_u, s_v\rangle/n - 1\}} \tag{49}$$

$$\overset{(i)}{\leq} \frac{4\sqrt{m}}{c^2} + 4\sqrt{\frac{c+1}{c^3}m\log\frac{2}{\delta}} + 2\sqrt{2m\log\frac{2}{\delta}} \tag{50}$$

where $(i)$ is from Pratap et al. [23] for reasonable values of $m$ and $\delta$ (Proof of Lemma 12). Gluing the above together, we prove the results. $c = 1$ leads to Theorem 3.2. $\qquad\square$

## A.3 Proof of Theorem 3.3

PROOF. We prove the existence of the MLP in the following by giving the recipe for constructing such an MLP and prove that it satisfies the conditions.

The MLP has an input layer of width $2n$, a hidden layer of width $N$, and an output layer of width 1. It can be formulated as

$$\text{MLP}(s_u, s_v) = \sigma\left(\begin{bmatrix} s_u & s_v \end{bmatrix}\mathbf{W}_1 + \mathbf{b}_1\right)\mathbf{W}_2 + \mathbf{b}_2 \tag{51}$$

where $\mathbf{W}_1 \in \mathbb{R}^{2n\times N}, \mathbf{W}_2 \in \mathbb{R}^{N\times 1}, \mathbf{b}_1 \in \mathbb{R}^N, \mathbf{b}_2 \in \mathbb{R}$, and $\sigma$ is the ReLU activation function.

We let

$$\mathbf{W}_1[i,j] = \begin{cases} 1 & \text{if } \mathcal{H}(v_j) = i \text{ or } \mathcal{H}(v_j) = i + N \\ 0 & \text{otherwise} \end{cases}, \mathbf{b}_1 = -1 \cdot \mathbf{1} \tag{52}$$

$$\mathbf{W}_2[i,0] = d(v_i), \mathbf{b}_2 = \mathbf{0} \tag{53}$$

The idea is that the hidden layer performs "set membership testing" for all $v \in \mathcal{V}$. Let $\mathbf{a} = \sigma\left(\begin{bmatrix} s_u & s_v \end{bmatrix}\mathbf{W}_1 + \mathbf{b}_1\right) \in \mathbb{R}^N$. Then, $\mathbf{a}_i = 1$ if and only if $v_i$ is a positive in both Bloom signatures $s_u$ and $s_v$, and $\mathbf{a}_i = 0$ if and only if $v_i$ is a negative in either Bloom signatures.

Consider expectation of the output of the MLP,

$$\mathbb{E}(\text{MLP}(s_u, s_v)) = \mathbb{E}\left(\mathbf{a}\mathbf{W}_2 + \mathbf{b}_2\right) \tag{54}$$

$$= \mathbb{E}\Bigg(\sum_{w_i\in\mathcal{N}(u)\cap\mathcal{N}(v)} d(w_i)\mathbf{a}_i + \sum_{w_j\in\mathcal{S}_C} d(w_j)\mathbf{a}_j$$

$$+ \sum_{w_k\in\mathcal{S}_{D_v}} d(w_k)\mathbf{a}_k + \sum_{w_l\in\mathcal{S}_{D_u}} d(w_l)\mathbf{a}_l\Bigg) \tag{55}$$

$$= S(u,v) + \mathbb{E}\Bigg(\sum_{w_j\in\mathcal{S}_C} d(w_j)\mathbf{a}_j\Bigg)$$

$$+ \mathbb{E}\Bigg(\sum_{w_k\in\mathcal{S}_{D_v}} d(w_k)\mathbf{a}_k\Bigg) + \mathbb{E}\Bigg(\sum_{w_l\in\mathcal{S}_{D_u}} d(w_l)\mathbf{a}_l\Bigg), \tag{56}$$

where $\mathcal{S}_C = \mathcal{V}\setminus(\mathcal{N}(u)\cup\mathcal{N}(v))$, $\mathcal{S}_{D_v} = \mathcal{N}(u)\setminus\mathcal{N}(v)$, and $\mathcal{S}_{D_u} = \mathcal{N}(v)\setminus\mathcal{N}(u)$.

Since Bloom filters have no false negatives, $w_i \in \mathcal{N}(u) \cap \mathcal{N}(v)$ implies $a_i = 1$. $\mathbf{a}_j$ for all $j$ such that $w_j \in \mathcal{V}\setminus(\mathcal{N}(u)\cup\mathcal{N}(v))$ are independent Poisson trials with $\Pr(\mathbf{a}_j = 1) = p_u p_v$, where $p_u = \left(1 - \frac{1}{n}\right)^{|\mathcal{N}(u)|}$ is the false positive rate for set membership testing of the Bloom signature of $u$, and similarly for $w_k \in \mathcal{N}(v)\setminus\mathcal{N}(u)$ and $w_l \in \mathcal{N}(u)\setminus\mathcal{N}(v)$.

By Chernoff bound, we obtain the three following inequalities,

$$\Pr\left(\sum_{w_j\in\mathcal{S}_C}\mathbf{a}_j \leq \left(1 + \sqrt{\frac{-3\log\delta}{|\mathcal{S}_C|p_u p_v}}\right)|\mathcal{S}_C|p_u p_v\right) \geq 1-\delta \tag{57}$$

$$\Pr\left(\sum_{w_k\in\mathcal{S}_{D_v}}\mathbf{a}_k \leq \left(1 + \sqrt{\frac{-3\log\delta}{|\mathcal{S}_{D_v}|p_v}}\right)|\mathcal{S}_{D_v}|p_v\right) \geq 1-\delta \tag{58}$$

$$\Pr\left(\sum_{w_l\in\mathcal{S}_{D_u}}\mathbf{a}_l \leq \left(1 + \sqrt{\frac{-3\log\delta}{|\mathcal{S}_{D_u}|p_u}}\right)|\mathcal{S}_{D_u}|p_u\right) \geq 1-\delta \tag{59}$$

Therefore, by Boole's inequality, we have

$$\Pr\Bigg(\text{MLP}(s_u, s_v) - S(u,v) \leq \left(1 + \sqrt{\frac{-3\log\delta}{|\mathcal{S}_C|p_u p_v}}\right)d_{\max}|\mathcal{S}_C|p_u p_v +$$

$$\left(1 + \sqrt{\frac{-3\log\delta}{|\mathcal{S}_{D_v}|p_v}}\right)d_{\max}|\mathcal{S}_{D_v}|p_v + \left(1 + \sqrt{\frac{-3\log\delta}{|\mathcal{S}_{D_u}|p_u}}\right)d_{\max}|\mathcal{S}_{D_u}|p_u\Bigg)$$

$$\geq 1 - 3\delta \tag{60}$$

$$\square$$

## A.4 Approximation of Neighborhood Union and Difference-based Heuristics

**Theorem A.6.** Suppose $S(u,v)$ is a neighborhood union-based heuristic with the maximum value of $d(\cdot)$ as $d_{\max} = \max_{w\in\mathcal{V}} d(w)$. Let $p_u = (1 - 1/n)^{|\mathcal{N}(u)|}$ denote the false positive rate of node $u$ for set membership testing in the Bloom signature $s_u$. Then, there

exists an MLP with one hidden layer of width $N$ and activation function $f(x) = \begin{cases} 0 & \text{if } x < 1 \\ 1 & \text{otherwise} \end{cases}$ where $x \in \mathbb{R}$, which takes the Bloom signatures of $u$ and $v$ as input and outputs $\hat{S}(u,v) = \text{MLP}(s_u, s_v)$, such that with probability 1, $\hat{S}(u,v) - S(u,v) \geq 0$; with probability at least $1 - \delta$, it holds that

$$\hat{S}(u,v) - S(u,v) \leq d_{\max}\left(1 + \sqrt{\frac{-3\log\delta}{|\mathcal{S}_C|p_u p_v}}\right)|\mathcal{S}_C|p_u p_v \quad (61)$$

where $\mathcal{S}_C = \mathcal{V} \setminus (\mathcal{N}(u) \cup \mathcal{N}(v))$.

PROOF. The proof is omitted since it is analogous to the proof of Theorem 3.3. □

**Theorem A.7.** Suppose $S(u,v)$ is a neighborhood difference-based heuristic with the maximum value of $d(\cdot)$ as $d_{\max} = \max_{w \in \mathcal{V}} d(w)$. Let $p_u = (1 - 1/n)^{|\mathcal{N}(u)|}$ denote the false positive rate of node $u$ for set membership testing in the Bloom signature $s_u$. Then, there

exists an MLP with one hidden layer of width $N$ and activation function $f(x) = \begin{cases} 1 & \text{if } x = 0 \\ 0 & \text{otherwise} \end{cases}$ where $x \in \mathbb{R}$, which takes the Bloom signatures of $u$ and $v$ as input and outputs $\hat{S}(u,v) = \text{MLP}(s_u, s_v)$, such that with probability at least $1 - 2\delta$, it holds that

$$\left|\hat{S}(u,v) - S(u,v)\right| \leq$$
$$\left(1 + \sqrt{\frac{-3\log\delta}{|S_{D_u}|p_v}}\right)d_{\max}|S_{D_u}|p_v + \left(1 + \sqrt{\frac{-3\log\delta}{|S_{D_v}|p_u}}\right)d_{\max}|S_{D_v}|p_u \quad (62)$$

where $\mathcal{S}_{D_v} = \mathcal{N}(u) \setminus \mathcal{N}(v)$, and $\mathcal{S}_{D_u} = \mathcal{N}(v) \setminus \mathcal{N}(u)$.

PROOF. The proof is omitted since it is analogous to the proof of Theorem 3.3. □

Received 20 February 2007; revised 12 March 2009; accepted 5 June 2009

