# OpenReview forum: "Learning Scalable Structural Representations for Link Prediction with Bloom Signatures"
_ACM.org/TheWebConf/2024/Conference — TheWebConf24_

### Official Review · Reviewer_U5Mc · 2023-11-19

**Novelty:** 4
**Technical Quality:** 5

**Review:**

This paper studies link prediction in graphs and propose to learn scalable structural representations with Bloom signatures. Compared with existing hash-based methods for link prediction, the proposed method is more efficient and needs less computational storage. The paper conducts extensive experiments to show the effectiveness and efficiency of the proposed method.

Strength:
1. The paper is overall well written and easy to understand.
2. Encoding node neighborhood with Bloom hashing is theoretically shown to be able to accurately approximate pairwise heuristics and learn pairwise relations in end-to-end manner.
3. The proposed method is shown to be effective in link prediction; it is also efficient and needs less storage.

Weakness:
1. For MPNN with Bloom signatures, the paper lacks theoretically illustration on its more expressiveness over vanilla GNNs.
2. Some important experiments are missing.
3. There also exist some typos and descriptive errors.

Details:
1. Adopting short and dense encoding for node neighborhood is useful in accurately approximating pairwise heuristics and learning pairwise relations in end-to-end manner. However, for MPNN with Bloom signatures, the paper lacks theoretical clarification. Currently, the paper only provides empirical explanation. In my eyes, the theoretical part can be derived correspondingly, which can further improve the overall quality of the paper.
2. The proposed method is easily affected by the selected hash function and the hash embedding size. The paper lacks the corresponding experiments. Further, which harsh function is selected in exp?
3. There exist some typos and descriptive errors. For example, in line 358, dense -> sparse; in line 552, in particularly -> particular; in line 664, consistently outperform SEAL -> this is not the truth; in line 536, to estimate to -> remove the second to.

**Questions:**

Please see the comments above.

**Reviewer Confidence:**

4: The reviewer is certain that the evaluation is correct and very familiar with the relevant literature

**Scope:**

4: The work is relevant to the Web and to the track, and is of broad interest to the community

---

### Official Review · Reviewer_vUVC · 2023-11-22

**Novelty:** 5
**Technical Quality:** 6

**Review:**

**Summary**

This paper addresses the challenge faced by graph neural networks (GNNs) in link prediction tasks, which require reasoning about pairs of nodes rather than individual nodes. The authors introduce an approach that augments message passing in GNNs with Bloom signatures—a hashing technique for node neighborhoods. This method effectively generalizes heuristics for neighborhood overlap estimation and shows promising results in link prediction tasks. Notably, it enhances scalability and computational efficiency, significantly outperforming baseline models in speed.

**Strengths**

- The proposed method generalizes existing heuristics for link prediction.
- The use of sparse hashing functions that can be pre-computed before training or inference increases the scalability and speed of the approach.
- Comprehensive experimental validation is conducted across datasets of varying sizes and types.
- In some instances, the performance of the proposed method surpasses traditional heuristic-based baselines, highlighting its versatility and effectiveness.

**Weaknesses**

- There is a large body of work on increasing the expressivity of GNNs, that is not only limited to the link prediction task. These works are especially interested in improving their performance with respect to the WL test, preserving equivariances, and higher-order structures (see for example [1, 2, 3]). It is not clear how the proposed Bloom Signatures heuristic compares with these approaches.
- The authors mention in their motivation use-cases such as knowledge graphs, and also citation networks in their experiments. These graphs are subject to frequent structural changes which for Bloom Signatures would require recomputing the signatures, but a discussion on this issue is missing.
- The selection of evaluation metrics appears inconsistent across datasets, making it challenging to conclusively determine the method's superiority. A uniform metric across all datasets or a comprehensive set of metrics for each would aid in a clearer comparative analysis.

**References**

[1] Maron et al. (2019), "Provably Powerful Graph Networks"

[2] de Haan et al. (2020), "Natural Graph Networks"

[3] Bodnar et al. (2021), "Weisfeiler and Lehman Go Topological: Message Passing Simplicial Networks"

**Questions:**

- How do Bloom Signatures compare with GNN architectures that are provably more powerful than standard GCNs in terms of heuristic benefits?
- Could you please elaborate on limitations of Bloom Signatures such new computations needed with structural changes (e.g. new nodes and edges added to the graph), and how they could be addressed?
- Would it be feasible to report a consistent set of metrics, or at least a single metric like AUC, across all datasets for a more uniform performance comparison against baselines?
- In line 206, you mention pairwise independent hash functions mapping to $[n]$. Could you clarify what $n$ refers to here, as it seems different from $N$, the number of nodes defined earlier?
- On line 410, Lemma 3.1 mentions the term 'sparsity $m$'. Could you provide a definition or elaboration on this term?

**Ethics Review Description:**

I do not see any ethical concerns with this work.

**Reviewer Confidence:**

3: The reviewer is confident but not certain that the evaluation is correct

**Scope:**

3: The work is somewhat relevant to the Web and to the track, and is of narrow interest to a sub-community

---

### Official Review · Reviewer_pEBf · 2023-11-22

**Novelty:** 3
**Technical Quality:** 4

**Review:**

The authors address the limitations of existing GNN methods, particularly in terms of pairwise feature computation for large-scale graphs and reliance on manual settings, in the context of link prediction on graphs. To overcome these challenges, they leverage the neighborhood information of nodes and propose Bloom Signature. This approach enhances the expressive power of traditional message-passing mechanisms by incorporating specifically designed structural features and provides improved scalability. Additionally, it adopts an offline preprocessing mode, making it well-suited for downstream tasks that require real-time performance, thus demonstrating practical applicability.

The text of this paper is well-written, and the content aligns with the aim and scope of WWW. The specific strengths and weaknesses are as follows.

**Strengths**

+ S1: Convincing motivation and impressive performance improvements.
+ S2: Scalability for traditional message-passing methods.
+ S3: Combination of offline preprocessing and online application mode.

Figure 1 effectively illustrates the challenges faced by widely used link prediction methods, providing a clear visual explanation for the performance improvements shown in Table 4. The framework design for pairwise features in Section 3.1 is specifically tailored to the link prediction task, thereby introducing missing information to existing message-passing methods on graphs. Furthermore, the integration of Bloom Signature with existing methods that focus on node neighborhoods provides valuable insights for related research. The combination of offline preprocessing and online application modes accommodates different downstream tasks, offering practical applicability to this approach.

**Weaknesses**

- W1: Insufficient analysis and explanation of the data distribution.
- W2: Unclear presentation of the framework, leading to difficulties in replicating the experiments.
- W3: Inadequate analysis of the experimental results.

The authors propose that the Bloom Signature seems to improve performance across various data distribution patterns, but this is not entirely intuitive. The authors should provide further explanation or clarify the constraints and limitations faced by the Bloom Signature. Additionally, the authors have not included a conceptual diagram or pseudocode for the overall connection prediction framework. In Section 4.1, there is not a detailed enough explanation of the experimental setup, making it challenging to replicate the experiments solely based on the paper itself.

On the other hand, the authors mention that "any type of neighborhood overlap-based heuristic can be estimated by a neural network which takes Bloom signatures as input." However, there is insufficient analysis in the experimental section regarding whether this mechanism has corresponding model preferences or tends towards specific data distribution patterns.

**Questions:**

1. Why can Bloom Signature be applicable to different data distribution patterns? Are there any specific requirements or limitations on data distribution in its practical application?
2. The embedding and reconstruction process resembles an autoencoder. Is the framework proposed by the authors related to autoencoders? What is the fundamental difference between them?

**Reviewer Confidence:**

4: The reviewer is certain that the evaluation is correct and very familiar with the relevant literature

**Scope:**

4: The work is relevant to the Web and to the track, and is of broad interest to the community

---

### Official Review · Reviewer_723k · 2023-11-22

**Novelty:** 4
**Technical Quality:** 5

**Review:**

Summary

The paper proposes to store node neighborhood information with bloom filter and use it to reconstruct structural information to enhance the expressiveness of GNN for link prediction. The bloom filter is supposed to reconstruct the structural information with low error and reduce the overhead during preprocessing, training, and inference.


Strength

S1. The paper provides details analysis to the computation cost and bloom filter error.

S2. The paper compares with state-of-the-art baselines.

S3. The idea to use bloom filter to store neighborhood information and use it to reconstruct structural information is interesting.


Weakness

W1. In Table 5, the paper only reports the  time cost on one dataset. But I think this table is as important as Table 4, which are both main experiment results of the paper. It should report the time cost over all datasets.

W2. The authors claim the method is 2 times faster than Buddy “under fair comparison”, which includes both inference and preprocessing time. But in real-world situations, online inference speed is usually more important than the pre-processing time cost, since the pre-processing can be done only once for the entire graph and be re-used many times. So if we only look at the inference speed, the proposed method is actually slower.

W3. The authors do not report the space overhead of the proposed method and the baselines. I noticed that in Figure 3, each node requires thousands of bytes as additional features, which could cause significant space overhead, prohibiting it to be applied to large graphs. Since the paper claims to be “scalable”, the space efficiency is also an important aspect.


W4. In Table 4, the results for OGBL-DDI is weird in that GCN, SAGE, and the proposed method have significantly higher accuracy than other methods. In particular, why is GAT significantly worse than GCN and SAGE? Why is the proposed method significantly better than other edge-wise baselines?

W5. The authors fail to discuss the limitation of the proposed method. In particular, what structural features cannot be constructed by the bloom filters. I believe at least the features related to distance cannot be constructed by it.

**Questions:**

See weakness above.

**Reviewer Confidence:**

4: The reviewer is certain that the evaluation is correct and very familiar with the relevant literature

**Scope:**

4: The work is relevant to the Web and to the track, and is of broad interest to the community

---

### Official Review · Reviewer_5wGD · 2023-11-26

**Novelty:** 5
**Technical Quality:** 5

**Review:**

Summary:
The paper introduces a hashing-based method named Bloom signatures to encode node neighborhoods efficiently. The proposed Bloom signatures can recover various types of edge-wise structural features, and can be integrated with the message passing framework of GNNs. Experiments demonstrated the effectiveness of the proposed method on OGB link prediction datasets.

Strengths:
+ The proposed approach of introducing bloom signature for better link prediction is novel.
+ Extensive experiments are performed to validate the effectiveness of the proposed approach.
+ Theoretical analysis is provided for the proposed approach. It seems solid.

Weaknesses:
To be honest, I am not an expert in hypergraph neural networks, while I have experience in applying GNNs, and familiar with the basic principle of GNNs. From my experience, the bloom-signatures-based approach proposed in this paper is very novel, and the technical details as well as the theoretical analysis of this paper seem solid.
- From Table 4, we can see the proposed approach outperforms other baselines significantly (such as in the ogbl-ddi dataset), while outperforms other baselines marginally. The reason should be analyzed in more depth.

**Questions:**

See the weakness.

**Reviewer Confidence:**

3: The reviewer is confident but not certain that the evaluation is correct

**Scope:**

4: The work is relevant to the Web and to the track, and is of broad interest to the community

---

### Decision · Program_Chairs · 2024-01-22

**Decision:**

Accept

**Comment:**

The paper presents a Bloom-signature based hashing approach to augment messaging passing in GNNs for link prediction. The approach obtains comparable performance to edge-wise GNNs for link prediction benchmarks with significantly better scaling performance. Reviewers found the choice of Bloom signatures for encoding node neighborhoods to be interesting, and the experiments are well designed and thorough. Sketching based approaches to encode node neighborhoods such as BUDDY have been proposed before, which limits the overall novelty of the proposed approach - however this paper compares against BUDDY and showcases better quality and lower preprocessing costs.